# Immunogenicity, Immune Dynamics, and Subsequent Response to the Booster Dose of Heterologous versus Homologous Prime-Boost Regimens with Adenoviral Vector and mRNA SARS-CoV-2 Vaccine among Liver Transplant Recipients: A Prospective Study

**DOI:** 10.3390/vaccines10122126

**Published:** 2022-12-12

**Authors:** Supachaya Sriphoosanaphan, Sirinporn Suksawatamnuay, Nunthiya Srisoonthorn, Nipaporn Siripon, Panarat Thaimai, Prooksa Ananchuensook, Kessarin Thanapirom, Bunthoon Nonthasoot, Pokrath Hansasuta, Piyawat Komolmit

**Affiliations:** 1Division of Gastroenterology, Department of Medicine, Faculty of Medicine, Chulalongkorn University, Bangkok 10330, Thailand; 2Centre of Excellence in Liver Diseases, King Chulalongkorn Memorial Hospital, Thai Red Cross Society, Bangkok 10330, Thailand; 3Liver Fibrosis and Cirrhosis Research Unit, Chulalongkorn University, Bangkok 10330, Thailand; 4Department of Surgery, Faculty of Medicine, Chulalongkorn University, Bangkok 10330, Thailand; 5Division of Virology, Department of Microbiology, Faculty of Medicine, Chulalongkorn University and King Chulalongkorn Memorial Hospital, Thai Red Cross Society, Bangkok 10330, Thailand

**Keywords:** heterologous, immunogenicity, liver transplant, SARS-CoV-2, vaccination

## Abstract

Background: Heterologous prime-boost vaccination potentially augments the immune response against SARS-CoV-2 in liver transplant (LT) recipients. We investigated immunogenicity induced by different primary prime-boost vaccination protocols and the subsequent response to the booster vaccine among LT recipients. Methods: LT recipients, who received primary immunisation with ChAdOx1/ChAdOx1 or ChAdOx1/BNT162b2, were administered the third dose of mRNA-1273 three months following the primary vaccination. Blood samples were collected before and after primary vaccination and post-booster. The levels of receptor binding domain antibody (anti-RBD) and neutralising antibody (sVNT) and spike-specific T-cell responses were assessed. Results: Among the 89 LT recipients, patients receiving ChAdOx1/BNT162b2 had significantly higher anti-RBD titres, sVNT, and cellular response after primary vaccination than those receiving ChAdOx1/ChAdOx1 (*p* < 0.05). The antibody response decreased 12 weeks after the primary vaccination. After the booster, humoral and cellular responses significantly improved, with comparable seroconversion rates between the heterologous and homologous groups. Positive sVNT against the wild type occurred in >90% of LT patients, with only 12.3% positive against the Omicron variant. Conclusions: ChAdOx1/BNT162b2 evoked a significantly higher immunological response than ChAdOx1/ChAdOx1 in LT recipients. The booster strategy substantially induced robust immunity against wild type in most patients but was less effective against the Omicron strain.

## 1. Introduction

The rapid spread of severe acute respiratory syndrome coronavirus-2 (SARS-CoV-2) has become a global public health concern and has been declared a pandemic since December 2019 [1]. Since the outbreak, a high mortality rate has been reported in vulnerable patients with comorbidities, including solid organ transplant recipients [2,3,4]. Patients who underwent liver transplant (LT) carry a higher risk of morbidity and mortality after SARS-CoV-2 infection, and this population should be prioritised for vaccination according to recommendations by professional societies [5,6].

Although considered safe and effective in minimising disease severity and mortality, several reports have revealed inadequate immune responses in LT recipients after a standard two-dose SARS-CoV-2 vaccination, with serological response rates ranging from 47–79% [7,8,9,10]. However, most studies are restricted to homologous vaccine regimens with messenger ribonucleic acid (mRNA)-based vaccines, whereas heterologous protocols have been less investigated. Heterologous vaccination is a potential strategy to improve SARS-CoV-2-specific immune response [11,12,13,14]. Heterologous immunisation with adenoviral-vector and mRNA vaccines induces more robust humoral and cellular immunity than homologous protocols in healthy individuals [13]. Nevertheless, information describing the immunogenicity of mix-and-match vaccine platforms in LT recipients is limited. Furthermore, most of the recently published data in this population focused mainly on humoral immune responses over a relatively short period [7,8,9,10]. The data available regarding cellular immune response, immune durability, and subsequent response to the booster dose in LT recipients are limited.

To explore these issues among LT recipients, we conducted this study with three main aims: (1) to evaluate SARS-CoV-2 specific humoral and cellular immune responses after primary vaccination with heterologous versus homologous prime-boost protocols, (2) to explore immune dynamics after primary immunisation, and (3) to assess the subsequent response to the booster vaccine following different primary vaccine series.

## 2. Materials and Methods

### 2.1. Study Population and Data Collection

This prospective, longitudinal observational study was conducted between June 2021 and February 2022 at a single liver transplant centre (King Chulalongkorn Memorial Hospital, Thai Red Cross Society, Bangkok, Thailand). SARS-CoV-2 naïve LT recipients aged ≥ 18 years were enrolled in the study. All LT patients were immunised with either homologous [ChAdOx1 (AstraZeneca, Cambridge, UK)/ChAdOx1] or heterologous [ChAdOx1/BNT162b2 (Pfizer Biotech, New York, NY, USA)] as their primary vaccine protocol. Vaccination schemes were determined by the national policies of the Ministry of Health of Thailand according to the availability of vaccines. All participants received an additional dose of mRNA-1273 (Moderna, Cambridge, MA, USA) three months following the standard two-dose vaccine series. The exclusion criteria were LT patients with a history of confirmed SARS-CoV-2 infection, prior immunisation with any SARS-CoV-2 vaccine, acute cellular rejection, or pregnancy. Blood samples were collected immediately before the first vaccination, four weeks after the second vaccination, twelve weeks after the second vaccination, and four weeks after the booster vaccine.

Demographic, clinical, and biochemical data were obtained from the electronic medical records at the time of the first vaccination. All LT recipients were screened for active respiratory tract infection and SARS-CoV-2 exposure by questionnaire at every visit. Adverse events after vaccination were assessed using a questionnaire and a phone interview.

### 2.2. Laboratory Assessment

#### 2.2.1. Anti-SARS-CoV-2 Antibodies

Anti-SARS-CoV-2 receptor-binding domain antibodies (anti-RBD) were measured by Elecsys^®^ anti-SARS-CoV-2 S using Cobas e411 immunoassay analysers (Roche Diagnostics, Rotkreuz, Switzerland). The sensitivity and specificity of this assay were 93.9% and 99.6%, respectively [15]. The results were reported as U/mL, equivalent to the binding antibody units (BAU)/mL [16]. The assay detection limit was 0.4 U/mL. We defined anti-RBD seroconversion as a cut-off of ≥132 U/mL, as suggested by the US FDA for the high titre for COVID-19 convalescent plasma [17].

#### 2.2.2. Surrogate SARS-CoV-2 Neutralising Antibodies

SARS-CoV-2 neutralising antibodies (NAs) were measured using a surrogate virus neutralisation test (sVNT) and cPass^TM^ NAs detection kits (GenScript, Piscataway, NJ, USA). This assay is a blocking ELISA detection tool that mimics the viral-host interaction. The protein-protein interaction between the horseradish peroxidase (HRP)-conjugated recombinant SARS-CoV-2 RBD fragment and human ACE2 receptor protein can be blocked by SARS-CoV-2 NAs. The sVNT results are reported as percentage inhibition. The cut-off for antibody detection was ≥ 30% inhibition, according to the manufacturer’s instructions. According to US FDA guidelines, the seroconversion rate of sVNT was defined as sVNT ≥ 68% inhibition [17]. In this study, we assessed NAs against the SARS-CoV-2 wild-type strain after primary and post-booster vaccinations. Given that the emergence of the Omicron variant has raised concern about the immune response against SARS-CoV-2, we also assessed sVNT to the Omicron strain in blood samples after the booster vaccine.

#### 2.2.3. T-Cell Response Assessment

SARS-CoV-2 specific T-cell responses were assessed using the enzyme-linked immunospot assay (ELISpot). In brief, ELISpot plates (Millipore, Watford, UK) were pre-coated with the human IFN-antibody (Mabtech, Nacka Strand, Sweden) at 4 °C overnight. Fresh peripheral blood mononuclear cells (PBMCs) were added in duplicate wells at 2 × 10^5^ cells in 50 μL per well and stimulated with SARS-CoV-2 spike peptide pools (S1 and S2) (Genscript, Piscataway, NJ, USA) in RPMI-1640 medium containing 10% FBS (Gibco, Waltham, MA, USA). Phytohemagglutinin (Sigma Aldrich, Burlington, MA, USA) and CMV lysates (Meridian Bioscience, Cincinnati, OH, USA) were used as positive and negative controls, respectively. The spots were counted using an ELISpot analyser (ImmunoSpot, Cleveland, OH, USA). Spot counts for negative control wells were subtracted from the test wells to quantify the intensity of the antigen-specific T-cell response. The results are presented as spot-forming units (SFU) per 10^6^ PBMCs. The mean value plus two standard deviations in the unstimulated wells was used as the lower limit to indicate a positive response.

### 2.3. Statistical Analyses

All statistical analyses were performed using the SPSS software (version 28.0; IBM Corp., Armonk, NY, USA), and scatter plots were generated using GraphPad Prism 8 (GraphPad Software, San Diego, CA, USA). Continuous variables were described as mean (SD) and median (IQR), as appropriate. Categorical data were described as frequencies and percentages. Anti-RBD titres, sVNT, and the number of IFN-secreting T-cells are expressed as the median and interquartile range (IQR). Chi-square and Fisher’s exact tests were used to compare the categorical variables. Student’s *t*-test or the Mann–Whitney U-test was used to assess differences between the groups. The impact of factors associated with immunological responses was analysed using a logistic regression model. Parameters with univariate *p*-values < 0.1 were included in multivariate analysis using the Spearman correlation test and presented as odds ratios (OR) with 95% confidence intervals (CI). The correlation between anti-RBD antibodies and other immune responses was assessed using Spearman’s rank test. Statistical significance was set at *p* < 0.05 unless otherwise stated.

### 2.4. Ethical Considerations

The study was reviewed and approved by the Ethics Committee and Institutional Review Board (IRB) of Chulalongkorn University, Bangkok, Thailand, and was performed in accordance with the Declaration of Helsinki (1989) of the World Medical Association (IRB number: 482/64). This study was registered with the Thai Clinical Trials Registry (TCTR) based on World Health Organization criteria (TCTR20210526004). All LT patients provided written informed consent for participation and publication.

## 3. Results

### 3.1. Baseline Characteristics

In total, 89 LT recipients were enrolled in the study. Sixty-four (71.9%) LT patients received a heterologous (ChAdOx1/BNT162b2) prime-boost vaccination protocol, whereas 25 (28.1%) participants received a homologous (ChAdOx1/ChAdOx1) vaccine regimen. mRNA-1273 was administered as the booster vaccine to all LT recipients three months following the standard two-dose vaccination (Appendix A). The clinical characteristics of the patients are summarised in Table 1. The mean age of the LT recipients was 57.8 ± 14.2 years, among which 68.5% were men. The median (IQR) time from LT to the first vaccination was 5.7 (2.8–11.8) years. Most patients underwent transplantation for viral aetiology, among whom 33.7% and 21.3% had chronic hepatitis B and C infection, respectively. Approximately 40% of the patients had hepatocellular carcinoma (HCC) at the time of transplantation. Hypertension (40.4%), diabetes mellitus (39.3%), and dyslipidaemia (44.9%) were common comorbidities among the LT patients in the study cohort.

Most LT recipients (86.5%) were treated with a combination of immunosuppressive therapies; 77.5% and 9.0% received two and three immunosuppressive agents, respectively. Calcineurin inhibitors (CNIs) were used as the backbone of the immunosuppressive regimen in 82.0% of patients. Mycophenolate mofetil (MMF) was administered to 55.1% of LT recipients, whereas 22.5% of patients received mammalian target-of-rapamycin (mTOR) inhibitors as immunosuppressive medications. The median CNIs serum levels, mTOR serum levels, and daily MMF doses are listed in Table 1.

The median time between the first and second vaccinations was 84 (78–86) days, and the median time between the second vaccination and the booster was 86 (81–94) days. No significant differences were observed in the demographic, clinical, and biochemical data between the LT recipient groups. All LT patients had a stable graft function before vaccination.

### 3.2. SARS-CoV-2 Specific Humoral Response

#### 3.2.1. Anti-SARS-CoV-2 RBD Antibodies and Post-Vaccination Antibody Kinetics

The SARS-CoV-2 specific humoral response was evaluated at four time points: immediately before the first vaccination, four weeks after the second vaccination, twelve weeks after the second vaccination, and four weeks after the booster. All LT recipients had negative anti-RBD IgG levels prior to the first vaccination.

Four weeks following the primary vaccination, the median anti-RBD titre was significantly higher in the ChAdOx1/BNT162b2 group compared with that in the ChAdOx1/ChAdOx1 group [842.9 (34.3–1884.0) U/mL vs. 152.1 (13.6–678.8) U/mL, (*p* = 0.02)] (Table 2). However, seroconversion occurred in 68.8% (44/64) of LT recipients receiving the ChAdOx1/BNT162b2 vaccine and 56.0% (14/25) of those receiving the ChAdOx1/ChAdOx1 vaccine; this difference was not statistically significant (*p* = 0.32) (Figure 1).

Twelve weeks after the primary vaccination, a decline in anti-RBD antibodies was observed in both primary vaccine platforms (Table 2). The median anti-RBD titre in the ChAdOx1/ChAdOx1 group decreased to 83.8 (14.2–312.3) U/mL (*p* = 0.09, 43.8% reduction rate), whereas the median anti-RBD IgG in patients receiving ChAdOx1/BNT162b2 significantly decreased to 638.5 (142.5–1245.5) U/mL (*p* < 0.001, 34.5% reduction rate). Nevertheless, the SARS-CoV-2 spike total antibodies remained significantly higher in the heterologous prime-boost group than in the homologous prime-boost group (*p* < 0.001).

The booster vaccine significantly improved immune response (Figure 2 and Table 2). The median anti-RBD titre significantly increased to 5134 (852.1–12,352.8) U/mL in the ChAdOx1/ChAdOx1 group and 10,346.0 (4889.0–15,298.5) U/mL in the ChAdOx1/BNT162b2 group (*p* < 0.001). Seroconversion occurred in 81.3% of LT recipients receiving ChAdOx1/ChAdOx1/mRNA-1273 and in 94.7% of those receiving ChAdOx1/BNT162b2/mRNA-1273. Although the additional vaccine yielded strikingly high antibody levels, no significant difference was observed in the anti-RBD titre and seroconversion rate between the two groups after the booster (*p* = 0.18 and *p* = 0.12, respectively) (Figure 1 and Table 2).

Potential variables as predictors of poor humoral response after primary vaccination were evaluated using logistic regression analysis (Table 3). In univariate analysis, the factors associated with the lack of humoral response were LT duration, tacrolimus serum level, MMF > 500 mg/d, and the use of an mTOR inhibitor. In the multivariate analysis, a daily MMF dose of >500 mg/d was the only independent prognostic factor for an impaired SARS-CoV-2 humoral response (odds ratio 21.30 [1.46–311.05], *p* = 0.025). Owing to the high seroconversion rate after booster vaccination, we could not identify significant predictors of the immune response following the third vaccine.

#### 3.2.2. Neutralising Antibody against SARS-CoV-2

The sVNT to the ancestral type (wild-type strain) was investigated at four weeks following primary immunisation and four weeks after the booster vaccination. Following primary immunisation, participants receiving ChAdOx1/BNT162b2 had significantly higher median sVNT levels than those receiving ChAdOx1/ChAdOx1 [91.2% (44.7–96.6%) vs. 39.9% (23.3–75.7%), *p* = 0.01] (Table 2). ChAdOx1/BNT162b2 vaccination also induced significantly higher seroconversion than ChAdOx1/ChAdOx1 vaccination (69.8% vs. 28.0%, *p* < 0.001) (Figure 1).

After an additional dose of mRNA-1273, the median sVNT level to wild type increased to 97.6% (96.8–97.8%) in patients vaccinated with ChAdOx1/BNT162b2 and 97.3% (69.2–97.6%) in those immunised with ChAdOx1/ChAdOx1; the difference was not statistically significant (*p* = 0.06) (Table 2). However, ChAdOx1/BNT162b2/mRNA-1273 elicited a significantly higher seroconversion rate among LT recipients than ChAdOx1/ChAdOx1/mRNA-1273 (*p* = 0.04). Notably, the sVNT level demonstrated a significant correlation with the anti-RBG titre after primary vaccination (r = 0.62, *p* < 0.001) and post-booster (r = 0.30, *p* = 0.03).

Neutralising antibodies to the Omicron variant were assessed in blood samples after booster vaccination. Notably, almost all LT patients had a substantially low level of sVNT to the Omicron strain (Figure 3). Only 10.5% of patients receiving ChAdOx1/BNT162b2/mRNA-1273 and 18.8% of patients receiving ChAdOx1/ChAdOx1/mRNA-1273 exhibited positive NAs responses to the Omicron strain.

### 3.3. SARS-CoV-2 Specific Cellular Response

The T-cell response was evaluated in 58 randomly selected LT patients: 41 in the ChAdOx1/BNT162b2 group and 17 in the ChAdOx1/ChAdOx1 group. The baseline clinical and laboratory characteristics were comparable between the two groups (*p* > 0.05) (Appendix A). The results of the IFN-γ ELISpot assay using pools S1 and S2 are shown in Figure 4. The spike-specific cellular response followed a trend similar to that of the humoral response. The percentages of positive responses after primary vaccination and booster were 51.2% and 80.5% in the ChAdOx1/BNT162b2 group and 23.5% and 52.9% in the ChAdOx1/ChAdOx1 groups, respectively. The median SFU/10^6^ PBMC cells of LT patients receiving the heterologous primary vaccine protocol was significantly higher than those receiving the homologous protocol (Table 2 and Figure 4). A significant correlation was observed between the anti-RBD titre (r = 0.32, *p* = 0.02) or sVNT (r = 0.30, *p* = 0.03) and spike-specific T-cell responses.

### 3.4. Safety

Overall, most patients experienced minor adverse events after injection of the primary vaccination and booster (Appendix A). Both vaccine regimens were well-tolerated. No graft rejection or severe AEs were noted. The most frequently reported adverse events were injection site pain, fever, and headaches. Comorbidities, vaccine platform, and immune response level were not associated with the development of adverse effects.

## 4. Discussion

In this prospective study, we evaluated the immunogenicity and dynamics of SARS-CoV-2 specific immune response as well as the subsequent benefit of the booster among LT recipients who received homologous versus heterologous primary vaccination. Following primary immunisation, the heterologous protocol (ChAdOx1/BNT162b2) induced a more robust immune response than that with the homologous protocol (ChAdOx1/ChAdOx1). The antibody level eventually decreased three months later, regardless of the vaccine platform. The additional third dose significantly improved humoral and cellular immune responses in both the primary vaccine groups. Most LT recipients had strong immunity against the wild-type strains. Nevertheless, the neutralising activity against Omicron variants was substantially attenuated in both groups, even after booster vaccination.

At the beginning of the pandemic, the mRNA vaccine was unavailable in most low- and middle-income countries. In Thailand, inactivated and adenoviral-vectored vaccines were the first vaccines to be inoculated in the general population during the outbreak before the arrival of mRNA vaccines [18]. The use of a heterologous immunisation protocol helped simplify the issues of vaccine shortage in Thailand, allowing us to explore the immunogenicity of different vaccine protocols among LT recipients. Our study suggested that the humoral response rates after primary vaccination were 56.0% and 68.8% in LT patients receiving homologous and heterologous protocols, respectively. Although not statistically significant, the heterologous vaccine scheme achieved higher antibody titres than in the homologous regimen. The antibody seroconversion rate in our cohort was in line with recently published reports on LT recipients immunised with the mRNA vaccine. Reuther et al. demonstrated that heterologous vaccination with the ChAdOx1/mRNA vaccine triggered stronger humoral and cellular immune responses than the homologous mRNA vaccine protocol in LT recipients [19]. However, this study included only a small number of LT patients receiving the ChAdOx1/mRNA vaccine (*n* = 11). In addition, Mendizabal et al. reported that heterologous adenoviral vector/mRNA improved the humoral response in Argentinian LT patients [20]. In our cohort, we confirmed the benefits of a heterologous primary vaccination scheme. Moreover, the important factor associated with impaired humoral response was a daily MMF dose of >500 mg/d, which confirms the role of MMF in the lack of vaccine response observed in LT recipients [8,21]. We also provided additional data regarding SARS-CoV-2 specific neutralising activity, which correlated with a more intense immune response.

We also evaluated cellular immune responses, which may contribute to immune longevity [22]. Our findings suggested that the heterologous protocol induced a significantly higher spike-specific T-cell response than that with homologous vaccination. The magnitude of the T-cell response substantially increased after the booster, which was in line with the results of the Israeli study [7]. Furthermore, the specific T-cell response is also correlated with the antibody response and neutralising activity.

Notably, the decrease in total anti-spike antibodies was observed in both vaccine protocols three months after the primary vaccination. Immune waning may diminish the protective effect against SARS-CoV-2, especially in patients in the homologous group, wherein less than half of those demonstrated detectable antibodies. Antibody kinetics were consistent with the results observed in a recent study in LT recipients [7].

Regarding the diminished immune response in LT recipients after primary immunisation [23,24], the World Health Organization has recommended an additional dose to extend primary vaccination for this high-risk population [25]. In this study, we confirmed the utility of a booster vaccine in enhancing the immune response in this population. The third dose improved the humoral immune response in more than 80% of LT recipients, regardless of the primary vaccine regimen, consistent with the study by Davidov et al. [7]. Moreover, we demonstrated that the booster dose also evoked neutralising activity and cellular response in most LT patients.

The emergence of variants of concern, for example the Omicron variant (B.1/B.1.1.529), has raised concerns about vaccine efficacy. With highly prevalent mutations in the spike protein resulting in the potential to escape the host immune response, it is anticipated that a markedly low humoral immune response against the Omicron strain will be observed [26]. Despite the significantly high level of neutralising antibodies against the ancestral type following the booster, our study confirmed that the additional vaccine elicited considerably poor neutralisation to Omicron, with less than 20% of immunised LT recipients achieving seroconversion. This result highlights additional strategies to protect against Omicron in vulnerable patients.

The strength of our study is that it was a longitudinal prospective study that provided comprehensive information regarding SARS-CoV-2-specific immune response, including anti-spike antibodies, neutralising activities, and T-cell responses, from baseline to post-booster in LT recipients. We investigated different types of primary vaccine regimens based on previous studies conducted in the Western region. Therefore, our results could be helpful in settings where the mRNA vaccine was not the predominant platform for primary immunisation or experienced vaccine supply issues, especially in developing countries.

Our study had some limitations. First, although we present clinical evidence in a real-world setting, the study was conducted in a single transplant centre, precluding generalisation to a broader population of LT recipients. Second, the study was limited by the absence of randomisation. We observed a relatively lower number of participants in the homologous vaccine group than in the heterologous vaccine regimen. However, the baseline characteristics were comparable between patients immunised with the two different vaccine platforms. Third, our study included only LT patients receiving homologous primary vaccination with ChAdOx1/ChAdOx1 and thus does not reflect whether ChAdOx1/BNT162b is superior to homologous BNT162b/BNT162b protocol. Lastly, there were no comparative data among healthy populations for immune responses. In a state of immunosuppression, weaker immune responses were expected among LT recipients in several studies [7,8,9]. Despite there not being a direct comparison, our cohort demonstrated a similar trend in immune dynamics and a significant increase in responses after the booster, as observed in immunocompetent individuals [27].

## 5. Conclusions

Primary vaccination with the heterologous regimen induced better SARS-CoV-2 specific immune response among LT patients than the homologous protocol. Waning immunity was observed after the primary immunisation, regardless of the vaccine regimen. The booster strategy produced a more robust immune reaction in most LT recipients. Nevertheless, protective efficacy against emerging virulent strains and immune sustainability after booster vaccination should be assessed in future studies.

## Figures and Tables

**Figure 1 vaccines-10-02126-f001:**
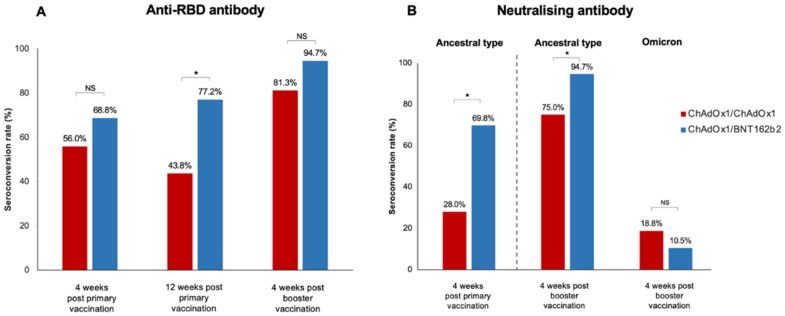
Seroconversion rate after primary vaccination and booster in LT recipients (**A**) Anti-SARS-CoV-2 receptor-binding-protein antibody (anti-RBD), (**B**) Neutralising antibody to the ancestral type and omicron strain after booster vaccination (*, statistically significant; NS, non-statistically significant).

**Figure 2 vaccines-10-02126-f002:**
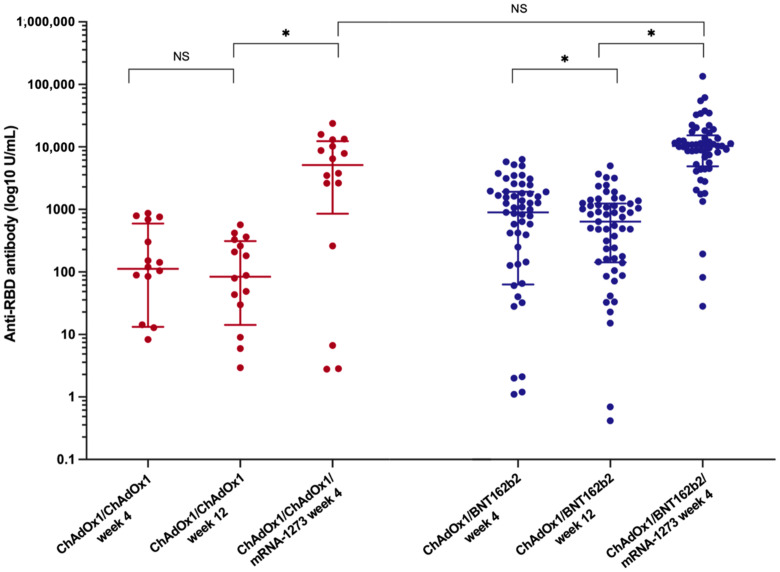
Dynamic of anti-SARS-CoV-2 binding-receptor-domain antibody (anti-RBD) after the primary vaccination and the booster vaccine in LT recipients among LT recipients (*, statistically significant; NS, non-statistically significant).

**Figure 3 vaccines-10-02126-f003:**
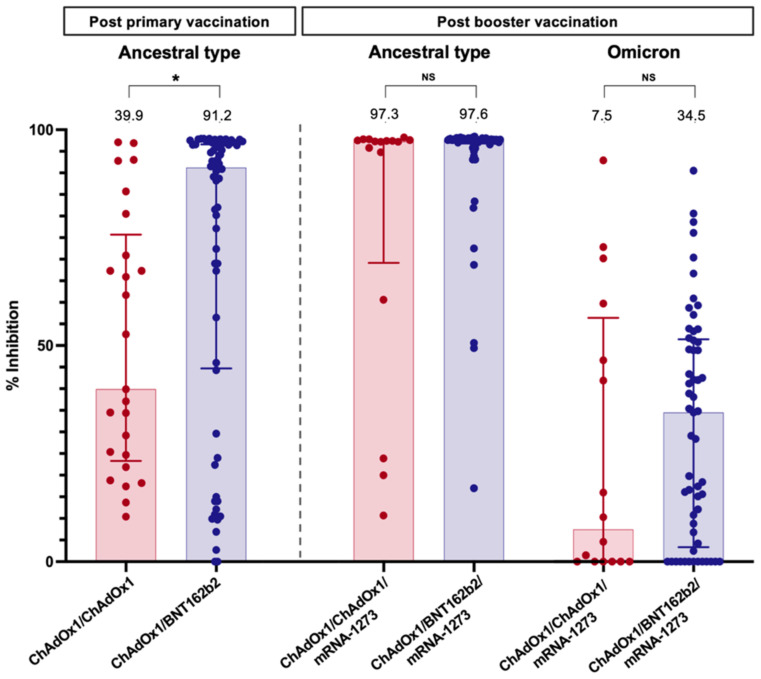
Neutralising antibody against the wild type (four weeks after primary vaccination and booster dose) and Omicron strain (four weeks after booster dose) in LT recipients. Values above bar graphs represent the median of % inhibition (*, statistically significant; NS, non-statistically significant).

**Figure 4 vaccines-10-02126-f004:**
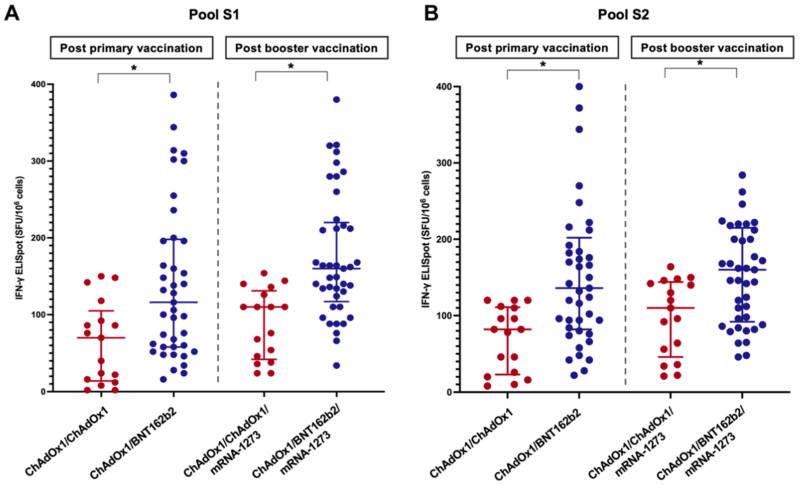
Cellular response to spike-specific T-cell responses among LT recipients four weeks following the primary immunisation and booster vaccine (**A**) Pool S1, (**B**) Pool S2 (*, statistically significant).

**Table 1 vaccines-10-02126-t001:** Baseline characteristics for LT recipients and comparison of LT recipients with heterologous and homologous prime-boost vaccination.

Parameter	All LT Recipients (*n* = 89)	ChAdOx1/BNT162b2 (*n* = 64)	ChAdOx1/ChAdOx1 (*n* = 25)	*p*-Value
Age (years)	57.8 ± 14.2	58.9 ± 12.7	55.1 ± 17.5	0.26
Sex, male (%)	61 (68.5)	41 (64.1)	20 (80.0)	0.21
BMI (kg/m^2^)	24.6 ± 4.1	26.2 ± 3.6	24.5 ± 5.1	0.93
Time after transplantation (years) *	5.7 (2.5–11.8)	5.7 (2.9–12.4)	5.1 (2.7–10.3)	0.89
Aetiology of liver disease (%)				
HBV	28 (33.7)	18 (28.1)	10 (40.0)	0.10
HCV	19 (21.3)	15 (23.4)	4 (16.0)	
Alcohol	15 (16.9)	12 (18.8)	3 (12.0)	
NASH	11 (12.4)	4 (6.3)	7 (28.0)	
Other	16 (18.0)	15 (23.4)	1 (4.0)	
HCC (%)	34 (38.2)	23 (35.9)	11 (44.0)	0.63
Comorbidity (%)				
HT	36 (40.4)	29 (45.3)	7 (28.0)	0.16
DM	35 (39.3)	21 (32.8)	14 (56.0)	0.55
DLP	40 (44.9)	30 (46.9)	10 (40.0)	0.64
CKD	16 (18.0)	9 (14.1)	7 (28.0)	0.14
Tacrolimus (%)	59 (66.3)	39 (60.9)	20 (80.0)	0.13
Drug level (ng/mL)	3.4 ± 1.8			
Cyclosporine (%)	14 (15.7)	11 (17.2)	3 (12.0)	0.55
Drug level (ng/mL)	389.1 ± 240.2			
Mycophenolate mofetil (%)	49 (55.1)	38 (59.4)	11 (44.0)	0.24
Daily dose (mg) *	1000 (250–1000)			
Sirolimus (%)	15 (16.9)	12 (18.8)	3 (12.0)	0.45
Drug level (ng/mL)	5.1 ± 2.1			
Everolimus (%)	5 (5.6)	2 (3.1)	3 (12.0)	0.10
Drug level (ng/mL)	3.6 ± 1.1			
Prednisolone	9 (10.1)	8 (12.5)	1 (4.0)	0.23
Daily dose *	2.5 (2.5–5)			
Regimen (%)				
1 immunosuppressant	12 (13.5)	8 (12.5)	4 (16.0)	0.67
2 immunosuppressants	69 (77.5)	49 (75.0)	20 (80.0)	
3 immunosuppressants	8 (9.0)	7 (10.9)	1 (4.0)	
TB (mg/dL)	0.7 ± 0.4	0.8 ± 0.4	0.7 ± 0.2	0.33
DB (mg/dL)	0.3 ± 0.1	0.3 ± 0.2	0.3 ± 0.1	0.08
AST (U/L)	24.1 ± 9.1	24.7 ± 10.2	22.5 ± 5.4	0.31
ALT (U/L)	24.9 ± 16.0	25.9 ± 28.2	22.1 ± 7.8	0.31
ALP (U/L)	88.9 ± 58.4	93.8 ± 65.3	76.6 ± 32.8	0.21
Albumin (g/dL)	4.2 ± 0.4	4.2 ± 0.3	4.2 ± 0.4	0.72
Hemoglobin (g/dL)	13.5 ± 2.9	13.6 ± 3.3	13.5 ± 1.8	0.92
White blood cell (10^3^/ul)	5.9 ± 2.2	6.0 ± 2.4	5.5 ± 1.8	0.35
Platelet (10^3^/uL)	212.9 ± 84.5	217.1 ± 88.5	202.1 ± 73.9	0.46
Creatinine (mg/dL)	1.1 ± 0.5	1.1 ±0.4	1.3 ± 0.6	0.08
Creatinine clearance (mL/min)	72.8 ± 29.7	74.2 ± 29.9	69.3 ± 29.7	0.49

* Median (IQR); ALP, alkaline phosphatase; ALT, alanine aminotransferase; AST, aspartate aminotransferase; BMI, body mass index; CKD, chronic kidney disease; DB, direct bilirubin; DLP, dyslipidaemia; DM, diabetes mellitus; HBV, hepatitis B virus; HCC, hepatocellular carcinoma; HCV, hepatitis C virus; HT, hypertension; NASH, non-alcoholic steatohepatitis; TB, total bilirubin.

**Table 2 vaccines-10-02126-t002:** Immune response in LT recipients after primary vaccination with heterologous or homologous SARS-CoV-2 vaccine.

Median (IQR)	ChAdOx1/BNT162b	ChAdOx1/ChAdOx1	*p*-Value
**Anti-RBD antibody (U/mL)**			
Week 4 after primary vaccination	842.9 (34.3–1884.0)	152.1 (13.6–678.8)	0.02
Week 12 after primary vaccination	638.5 (142.5–1245.5)	83.8 (14.2–312.3)	<0.001
Week 4 after booster	10,346.0 (4889.0–15,298.5)	5134 (852.1–12,352.8)	0.18
**Neutralising antibody (% inhibition)**			
Week 4 after primary vaccination	91.2 (44.7–96.6)	39.9 (23.3–75.7)	0.01
Week 4 after booster	97.6 (96.8–97.8)	97.3 (69.2–97.6)	0.06
**Spike-specific T-cells response (SFU/10^6^ PBMC)**			
** *Pool S1* **			
4 weeks after primary vaccination	116 (58–198)	70 (14–105)	0.02
4 weeks after booster	160 (117–220)	110 (42–131)	0.004
** *Pool S2* **			
4 weeks after primary vaccination	136 (82–202)	82 (23–111)	0.02
4 weeks after booster	160 (92–215)	110 (46–144)	0.03

IQR, interquartile range; LT, liver transplant; PBMC, peripheral blood mononuclear cells; RBD, receptor binding domain; SFU, spot-forming unit.

**Table 3 vaccines-10-02126-t003:** Predictors of poor humoral immune response after primary vaccination in LT recipients *.

Parameter	Univariate OR	*p*-Value	Multivariate OR	*p*-Value
Age	1.03 (0.99–1.06)	0.084		
Sex (male)	0.84 (0.33–2.17)	0.718		
LT duration (year)	1.12 (1.01–1.23)	0.023	0.85 (0.61–1.18)	0.324
Vaccine regimen	0.58 (0.22–1.50)	0.259		
BMI	1.02 (0.92–1.14)	0.683		
HT	0.59 (0.24–1.46)	0.252		
DM	1.18 (0.49–2.86)	0.713		
Tacrolimus	1.38 (0.54–3.56)	0.500		
Tacrolimus level	0.66 (0.47–0.94)	0.020	0.62 (0.28–1.34)	0.227
MMF >500 mg/day	10.80 (2.54–45.87)	0.001	21.30 (1.46–311.05)	0.025
mTOR inhibitor	0.15 (0.03–0.71)	0.017	N/A	
Triple immunosuppression	2.00 (0.46–8.61)	0.352		
CrCL >60 mL/min	0.64 (0.26–1.56)	0.325		

* Poor humoral immune response was defined as anti-RBD <132 U/mL. BMI, body mass index; CrCL, creatinine clearance; DM, diabetes mellitus; HT, hypertension; LT, liver transplant; MMF, mycophenolate mofetil; mTOR, mammalian target-of-rapamycin; OR, odd ratio.

## Data Availability

The datasets generated and analysed during the current study are available from the corresponding author upon reasonable request.

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
