# Peer review of "Immunogenicity, Immune Dynamics, and Subsequent Response to the Booster Dose of Heterologous versus Homologous Prime-Boost Regimens with Adenoviral Vector and mRNA SARS-CoV-2 Vaccine among Liver Transplant Recipients: A Prospective Study"

_vaccines, 2022, doi:10.3390/vaccines10122126_

Round 1

Reviewer 1 Report

The paper is interesting and well written. Is well structured, statystical analysis are adequate, methology is correct and consistent with endpoint of the study. Results are described well. Figures are good and exhaustive. I suggest only tu briefly discuss the importance of vaccinations in autoimmune disease for confirming the need of vaccination in frialty subjects (see and add as references papers by Murdaca et al concerning this topic)

Reviewer 2 Report

The authors examined immune response in LT patients vaccinated with three SARS-CoV-2 vaccines and found that ChAdOx1/BNT162b2 elicited significantly higher humoral and cellular responses than ChAdOx1/ChAdOx1. Booster dose of mRNA-1273 induced substantial immunity against wild type in most patients, but were less effective against Omicron strain.

Studies of SARS-CoV-2 vaccines for these immunocompromised patients who have received LT will be necessary. Therefore, this study is valuable.

However, there are some points to reconsider, as shown below.

From the title of this article, the authors seem to reveal that homologous vaccination is inferior to heterologous primary and booster regimens in immune response in patients with LT. However, this study did not include homologous vaccine group with BNT162b2. Therefore, it is unclear whether ChAdOx1/BNT162b2 is superior to BNT162b2/BNT162b2. Also, do the present results mean that mRNA vaccines are more effective than adenovirus-based vaccines? These points should be explained.

The name of the booster vaccine, mRNA-1273, should be described in the Abstract.

Surrogate SARS-CoV-2 neutralizing antibodies: NT is the most important serological index for estimating the efficacy of each vaccine to prevent SARS-CoV-2 infection. The authors used sVNT to assess NT in vaccinated patients, but the inhibition value was 100% after mRNA-1273 booster and no precise value was obtained (Fig 3). NT titers have been measured in other previous studies (ref. 6, 12, 22). Without quantifying NT, the actual efficacy of each vaccination cannot be compared. The authors need comments on this.

Table 1: Lines for Cyclosporine, Mycophenolate mofetil, --- Prednisolone are skewed. This should be rectified.

Figure 1: It is unclear which bar represents the heterologous and homologous vaccine groups. A and B should also be shown in the Figure.

Table 3: Criteria should be explained how patients were classified into the lack of humoral response group. The risk of LT recipients with a low humoral immune response after the second vaccination was associated with age, arterial hypertension, and calcineurin inhibitor, according to ref. 18. The clinical indicators associated with poor humeral immune response should be discussed.

Figure 3: Weeks post-vaccination should be listed in the figure legend.

Figure 4: Weeks post-vaccination should be listed in the figure legend.

Line 285, line 287: heterologous protocol (ChAdOx1/ ChAdOx1) should be heterologous protocol (ChAdOx1/BNT162b2). Similarly, homologous protocol (ChAdOx1/ BNT162b2) should be homologous protocol (ChAdOx1/ ChAdOx1).

Line 322: The authors stated that “the decay of total anti-spike antibodies was observed in both protocols three months after the primary vaccination”. “Decay” may be “decrease”.

Line 332: The authors stated that “more than 90% of LT recipients.” Shouldn’t it be 81.3% and 94.7%?

A similar study has been reported in healthy individuals from the same country (ref. 23). Data from healthy controls would be used for comparison to discuss how LT affects the efficiency of SARS-CoV-2 vaccination.

Reviewer 3 Report

Comments to Authors 

            This study showed that a) primary vaccination with the heterologous regimen induced better SARS-CoV-2 specific immune response among LT patients than the homologous protocol; b) waning immunity was observed after the primary immunisation, regardless of the vaccine regimen; c) the booster strategy produced a more robust immune reaction in most LT recipients; d) nevertheless, protective efficacy against emerging virulent strains and immune sustainability after booster vaccination should be assessed in future studies.

           Authors are kindly requested to emphasize the current concepts about these issues in the context of recent knowledge and the available literature. This articles should be quoted in the References list.

References

11.      Humoral and cellular response of COVID-19 vaccine among solid organ transplant recipients: A systematic review and meta-analysis [published online ahead of print, 2022 Aug 4]. Transpl Infect Dis. 2022;e13926. doi:10.1111/tid.13926.

22.      Immunogenicity of an mRNA-Based COVID-19 Vaccine among Adolescents with Obesity or Liver Transplants. Vaccines (Basel). 2022; 10 (11): 1867. Published 2022 Nov 4. doi:10.3390/vaccines10111867.
